# The Influence of Marginalization on Cultural Attitudes and Trash Disposal Practices in Esfuerzo de Paraíso of the Dominican Republic: A Qualitative Interview Study

**DOI:** 10.3390/ijerph18062872

**Published:** 2021-03-11

**Authors:** Madison Sasman, Carrie B. Dolan, Daniel Villegas, Estelle Eyob, Catherine Barrett

**Affiliations:** Department of Health Sciences, William & Mary, Williamsburg, VA 23185, USA; mcsasman@email.wm.edu (M.S.); cbdolan@wm.edu (C.B.D.); ndvillegas@email.wm.edu (D.V.); eeyob@email.wm.edu (E.E.)

**Keywords:** public health, waste management, burning trash, qualitative methods, marginalized communities

## Abstract

The inadequate management of municipal solid waste (MSW) in fast-developing nations is a major public health problem. Trash collection is often inconsistent, leaving residents to use unsafe disposal methods such as incineration or unregulated dumping. The issue is especially pronounced in marginalized communities, where public service provision is scarce. Past research has identified factors that perpetuate harmful disposal practices. The current study expanded on previous work by exploring how individuals’ perceptions of political, spatial, and economic marginalization affected their agency with regards to waste management. Researchers focused on a marginalized community in the Dominican Republic known as Esfuerzo de Paraíso. There, they conducted semi-structured interviews to explore residents’ perceptions of marginalization at the individual, interpersonal, community, and institutional levels, and its effects on their agency. A qualitative coding process revealed that most community members were discontent with their trash disposal practices, but that long-standing marginalization left them feeling ill equipped to generate change at the individual level. Interviewees believed that change should be initiated at the community level and implemented with the support of institutional-level actors, namely the municipal government. Residents did not identify any non-governmental organizations as possible sources of help, which may suggest a limited view of institutional support networks.

## 1. Introduction

### 1.1. Literature Review

The inaccessibility of waste management services is a prominent concern for low-income countries. Many low-income countries (LICs) have inadequate waste management systems due to a lack of awareness, technology, finances, and policy governance [1]. In addition, LICs are constrained by a lack of economic, spatial, and educational resources. [2]. These countries have limited land to operate proper recycling and trash disposal practices, on top of limited funding for transportation and maintenance. Therefore, alternate waste management practices such as incineration and unregulated dumping are commonly used systems in these countries [2].

There is a robust literature that establishes a connection between a lack of frequent and effective public sanitation provisions and negative physical or environmental health outcomes in LICs [2,3,4,5]. It was determined that in LICs, less than 50 percent of all municipal solid waste (MSW) was properly collected, whereas in high-income countries, 98% of MSW was properly collected [5]. Furthermore, MSW exposure in LICs was linked to negative health outcomes, although it is difficult to quantify long-term health impacts due to lack of data [5].

MSW causes negative health outcomes independent of environmental pollution [3]. A study conducted by Ferronato and Torretta found that open-dumping practices led to the spread of infectious pathogens through disease vectors [3]. Since dumping grounds were not properly contained, the resulting leachate also exposed high-density populations to toxic waste [3]. Additionally, the open burning of MSW directly led to respiratory illnesses in these communities [3]. Improper trash collection also contributes to pollutants and environmental health problems. A study in Ecuador found that the accumulation of unregulated waste created leachate, which leaked toxic liquid into the ecosystem and created a breeding ground for bacteria and parasites [6]. The improper disposal of MSW can also wreak havoc on sewer systems, leading to water pollution that affects all aspects of the ecosphere [7]. The practice of open burning contributes to the production of greenhouse gases (GHG) such as carbon monoxide, carbon dioxide, and nitric oxide [3]. Although not as widely mentioned, landfills and open-dumping grounds are the third leading source of methane gas released into the environment [8]. Without proper waste management services, masses of recyclable materials end up in dumping grounds; consequently, they contribute to GHG emissions and have negative long-term environmental health implications [8]. Despite advances towards implementing solutions to waste disposal problems, a variety of operational, socioeconomic, informational, and regulatory barriers prevent lower-income countries from making trash disposal more accessible and sustainable [9,10].

As the literature above has stated, marginalized communities experience negative health and environmental outcomes due to ineffective governance and inadequate resources. In low-income countries governmental resources are inadequate for the majority of the population; however, marginalized communities are disproportionately affected by the lack of access to resources [11,12]. Therefore, the most drastic health outcomes will reflect in the most vulnerable communities [13]. To resolve the issue of the inaccessibility of waste management services, one must also focus on resolving these disparities.

A study conducted by Burrell, Song and Clements found that residents of rural villages in Honduras were likely to dispose of their trash through incineration and open-air dumping [14]. Furthermore, the villagers believed that the trash accumulation acted as both an eyesore and a public health hazard. The study found low support for an incentive program related to safe trash disposal practices, with only 1.2 percent of respondents supporting the idea. More popular ideas included increased education of proper trash disposal practices and community-wide cleaning organizations. Despite these findings, the study failed to address obstacles that originated outside of the community sphere and perpetuated improper waste management practices; for example, a lack of consistent trash collection services that would afford residents the opportunity to dispose of their trash safely and sustainably. Furthermore, while it did use community member perceptions as a primary data source, it did not consider the underlying marginalization that affects these perceptions. Part of the reason for this was a methodological focus on quantitative survey statistics, rather than a qualitative analysis of individual and community perspectives. Another case study by Tadesse, Ruijis, and Hagos in Mekelle, Ethiopia, examined the impact of socioeconomic variables, waste facilities, and environmental concerns on residents’ waste disposal methods [15]. They reported that the main factors affecting waste management practices were spatial and economic, since those who lived farther away from communal waste containers and those with lower incomes were more likely to use open dumping [15].

Researchers focused on household-level variables, but they also called for further research into the ways that institutional variables could shape the larger household waste management system [15]. Similar to the previous study in Honduras, the findings did not consider how residents’ perceptions of marginalization and agency, not only at the individual level but also at the institutional level, could have determined impactful solutions. In fact, in a similar study Ferronato et al. stated the need for integration of private-public partnerships as well as intervention of non-governmental organizations (NGOs) in order to improve the MSW management of La Paz, Bolivia [16] (p. 297). Although this would be a good starting point, it is not enough to utilize outside resources without gathering the input of affected communities.

According to Jordan and Tuijl, it is important for NGOs dealing with international development to move past positions of advocacy [17]. Interventions, when conducted through cross-cultural collaborations, should first involve a period of social learning for foreign researchers [17]. This supports the findings of Snyder et al. who discovered that community buy in is difficult to obtain when researchers are constrained to a short time frame [18]. Not having an established relationship with the partner community leads to a lack of trust when attempting to enact positive change. In this light, it is important to consider the social dynamics of a community in addition to its resources [19].

There is a significant gap in the literature concerning the effect of community-level perceptions of marginalization on waste-management practices and beliefs [20]. Addressing MSW practices at higher socioecological levels by seeking partnerships at the community and institutional levels is critical to combating all levels of marginalization and creating sustainable outcomes [21,22]. Therefore, this research project uses a socioecological model to provide an initial inquiry on how social, political, and spatial marginalization affect a community’s agency in terms of the waste management practices they employ, especially when the community acknowledges the negative public health effects of these practices. To answer this question, we conducted a qualitative interview-based study in Esfuerzo de Paraíso, a marginalized community in the Dominican Republic. Through this study, we aimed to deepen our understanding of the conditions that result in a gap between the community’s preferred waste management methods and the actual practices that are utilized.

### 1.2. Conceptual Framework

We draw upon a socioecological model that focuses on considering how individuals, communities and societies are intertwined. Our specific interest is the relationship community members of Esfuerzo have with their natural, social and political environments. We use interdisciplinary methods to demonstrate the overlapping sources of influence on community members’ practices towards trash disposal and the subsequent implications on their health. We seek to engage in multiple levels of influence simultaneously to address problems that arise at multiple levels. Through this framework, we seek to understand the importance of interventions taken at each level of influence. By using the different levels of analysis, we can better clarify the differing values and goals for interventions at each level and ensure that proper solutions are implemented at the right level of influence.

## 2. Materials and Methods

### 2.1. Data Collection

Researchers conducted on-site interviews in Esfuerzo de Paraíso, Santo Domingo Norte, Dominican Republic, in January 2020. The research consisted of twenty-seven semi-structured, in-depth interviews with community members which ranged from five to thirty-two minutes in length. The average interview lasted twelve minutes. A standard interview protocol was used for all participants (Table 1). This consisted of a semi-structured interview format in which all participants were asked a standard set of questions by the interviewers, and interviewees could direct the conversation as they wished in order to best obtain a quality response. Semi-structured interviews allowed for more flexibility in participant’s answers, while still providing a standardized format to which researchers could draw conclusions.

To identify participants, researchers utilized a random route procedure in the community of Esfuerzo. Based on a detailed map of the community, which was created by researchers in collaboration with community members, the community was divided into four block groups of approximately equal size (Figure 1). A random number generator was used to select a household in each block as a starting point. From that point, researchers interviewed every third household by counting doors along the right side of the street and turning right at all intersections. After reaching a dead end or the boundary of the block, they would turn around and begin counting on the other side of the street. This process was repeated until every house in each block had been either interviewed or counted over, in order to include every community household in the sampling frame. If a selected household had no adults present at the time of our initial visit, interviewers returned to the house later that day or on a subsequent day. If the household was still empty on the next day, the next house on the right was selected as a replacement. Interviews were conducted by two teams of three student researchers each. Both teams used the same interview guide consisting of twelve questions about waste management practices as well as eight questions about other community concerns that would inform our ongoing partnership with Esfuerzo. This research method best allowed researchers to highlight individual perspectives, while also gathering conclusions about the community as a whole.

### 2.2. Data Analysis

All of the interviews were recorded and de-identified. Audiotapes of the interviews were transcribed and translated using the software Sonix (Sonix, Inc., San Francisco, CA, USA). Audiotapes were then independently coded by two research team members using a coding sheet developed by the research team. The codes consisted of keywords and phrases used to organize the data into broad, overarching categories. NVivo, a qualitative analysis software program, was then used to group all responses by key themes in order to better understand community members’ feelings and experiences. The members of the research team reviewed the analysis to identify key themes. A total of five major themes were identified, each with their own set of nodes and subnodes: Community Engagement, Government, Health, Trash, and Water. These categories of codes provided answers to research questions. Researchers selected quotes from the interviews to emphasize the results and describe developing patterns of commonality. The final stage of the data analysis process was data verification, which involved rechecking the transcripts and codes for accuracy and consistency.

### 2.3. Research Ethics

Ethical approval for this research was provided by the affiliated university prior to the commencement of the study (PHSC-2019-12-13-14014-cbdolan). Participants gave consent prior to the study. Researchers made it clear to the participants that their participation throughout the research process was voluntary and they could withdraw at any point. The anonymity and confidentiality of the participants was preserved by not revealing their identity in the analysis and reporting of findings.

## 3. Results

This section provides an overview of the study sample and descriptive findings derived from the on-site interviews, with the intended outcome of better understanding the trash disposal methods in marginalized communities and how varying spheres of influence impact community members’ actions. The methods utilized in this research were qualitative and therefore do not attempt to establish statistical associations between waste disposal and health outcomes. These methods are useful to identify hypotheses that could be tested using quantitative measures. The results do not allow for any official conclusions.

### 3.1. Study Sample and Basic Demographics

Of the twenty-seven participants, the majority were female (*n* = 16). Each participant was 18 years of age or older, and each lived in the community of interest, Esfuerzo de Paraíso. Five interviewees resided in Block A, eight resided in Block B, seven resided in Block C, and seven resided in Block D. Table 2 and Table 3 present this demographic data.

### 3.2. Trash Disposal Methods

The principal ways that residents disposed of trash were by burning it or depositing it at an improvised dumping ground near the community. In total, 15 interviewees burned trash, 8 threw trash in the dump, 2 did a combination of both, and 1 interviewee reported sorting trash in order to burn paper and reuse plastic containers. Only one participant threw trash in the stream that runs alongside the community. This stream is referred to as the cañada, and it is often overflowing with garbage. There is broad consensus that most of the trash in the cañada runs down from a neighborhood located above Esfuerzo.

The residents who used the dump either brought trash to the dump themselves or paid others to transport it. This trash transportation service was an informal system that operated on an ad hoc basis. One resident said:

“There’s a kid..he’s...paid for it. [To] throw out the trash,...come and throw it away.”

Burning trash was viewed as slightly more convenient than depositing trash at the dump, because residents did not have to leave the neighborhood. One interviewee explained why she occasionally burned trash:

“I would say for the comfort of not going out...far away...Sometimes I burn it, but...we take it and throw it out.”

Within the sample, trash burning was the predominant method of waste disposal.

### 3.3. Health Implications of Trash Disposal Methods

#### 3.3.1. Physical Health

Interviewees referenced concerns about physical health 35 times across 19 different interviews. Some of their primary concerns were the negative health effects that resulted from frequent flooding of the trash-filled cañada. Residents felt that floods exposed them to contaminated water and an abundance of mosquitoes. When asked about factors that impacted health, one resident echoed others and observed that:

“...mosquitoes and water, that is... [a] problem of the cañada...The water is dirty and all the trash...leaches out...”

A few residents believed that trash accumulation in the cañada contributed to damaging floods. One interviewee explained that there were fewer floods during the times when the stream was free of garbage:

“...if they [the local government] worked on the dredging, cleaning,...that would be a step forward...because a couple of years ago, after cleaning up the cañada,...there [was] a time of silence.”

Aside from flood-related health concerns, residents identified subpar trash disposal methods as a source of harm to physical health. Many reported that the fumes from burning garbage exacerbated asthma, caused cough, and made it more difficult to breathe. One interviewee who burned trash said:

“It’s affecting health, mostly because here, burning them [trash piles] affects...my child,...my asthmatic mother. This is a problem.”

#### 3.3.2. Environmental Health

Concerns about environmental health were brought up 23 times across 12 interviews. Many interviewees believed that their trash disposal methods had a negative impact on the environment. One person expressed that:

“[Burning the garbage]...does harm. How bad! I know the environment doesn’t like that.”

Another community member acknowledged the long-term consequences of non-ideal trash management:

“It [the trash] always hurts [the environment], even if it’s not all at once. But...as time [passes it] hurts [the environment].”

### 3.4. Community Frustrations with Government Neglect and Consistent Marginalization

Community members expressed general frustration with the local government 50 times across 21 interviews. In 66% of these instances, residents conveyed a lack of trust in government officials. Most said that politicians neglected them and ignored their needs. One interviewee explained that:

“In this neighborhood they have it as the neighborhood of oblivion, because here... [The politicians] come, they promise and they promise. They do not follow-through. This neighborhood does not exist for the politicians...It’s always the same.”

Others said that politicians only paid attention to the community during campaign cycles. Several residents agreed with an interviewee who said that:

“Trucks come to pick up trash when it’s political time.”

At the time of research, residents reported that the sporadic provision of government-sponsored trash trucks never lasted for more than a few weeks. In total, 26 out of 27 interviewees brought up the lack of follow-through, and many talked about it at length. The issue was referenced 56 times in total. Residents noted that they did not have adequate resources to manage the trash on their own; one interviewee explained that it was not financially feasible for most community members to hire private trash-management services:

“...sometimes someone can’t afford 30 or 20 pesos.”

Another interviewee said that residents needed government support to manage the garbage that ran down into the cañada:

“...all the garbage accumulated there comes from up there.. from another neighborhood....let...the City Council send to collect it, because otherwise we can not.”

Finally, one respondent lamented that although the community would like to find ways to clean the cañada and prevent flooding:

“...we cannot because we do not have the instruments, we do not have the mechanisms, we do not have the money and we do not have the means. They [the government] does, because they have...their stuff to work with, but you know...we have a losing battle.”

In general, community members felt ill equipped to manage trash. They were discontent with their current options for trash disposal, as well as the steady accumulation of garbage in the cañada. Several residents believed that their problems were due to consistent political marginalization, but a few others also speculated that the isolated location of the community impeded adequate service provision. Esfuerzo de Paraíso sits at the bottom of a large hill and is set apart from other residences. The roads in the community are rough and unpaved. One interviewee asserted that:

“What are we going to do here? It’s kind of an isolated place...Maybe they [the trash collectors] wouldn’t come...”

A different resident expressed similar sentiments, saying:

“...they [the trash trucks] are not going to come down because they are not interested in that around here, [the terrain is] uncomfortable and somewhat awkward.”

Nevertheless, residents believed that their hard-to-reach location was no excuse for government neglect. An interviewee who was fed up with the community’s spatial marginalization, said:

“The service and the role of the City Council in waste disposal, as it should be, should be for everyone, not only for some part of the population, but it should be for everyone. They should go in all the places, pick up the trash.”

### 3.5. Community Engagement and Perceived Agency

Residents had many ideas about how to improve their situation, especially with regards to waste management. Community-provided suggestions appeared 60 times across 26 interviews. When residents were asked how trash management in the community could be improved, there was widespread agreement that the solution rested upon the steady provision of government-sponsored trash trucks. Many believed that members of the neighborhood board should lobby the city council and push for reliable trash collection. One interviewee said that:

“The relationship between the Community and the Town Hall should be serious. The Junta de Vecinos [the neighborhood board] should have to go there to call [for trash trucks]. It should be a system... I think the Junta de Vecinos may have a...contact,...I think it’s enough to make it happen.”

In general, residents also said that they would be willing to make individual contributions toward a solution. They made statements that signaled high community engagement 38 times across 21 interviews. In comparison, expressions of low community engagement appeared only 13 times across 6 interviews, with 7 of those comments sourced from 1 interview. The interviewees that displayed high community engagement had faith in their fellow neighbors and an enthusiasm for change. One interviewee said:

“...I think they [community members] would agree [to contribute to a solution], of course they would.”

Another stated:

“Well, I’m willing to do what you say, if it has to be done, if the street has to be cleaned, whatever...to keep it clean and I’ll do it.”

The few residents who displayed low community engagement felt as though they lacked agency and efficacy. One individual simply said:

“The only solution...is change and change does not happen...we can’t do anything.”

### 3.6. Geography-Based and Gender-Based Comparisons

In addition to analyzing the interviews both individually and collectively, we also analyzed the data by different attributes. Each interview was classified with two attributes: geographical block and gender. The purpose of this method was to make comparisons between groups and see if there were notable similarities or differences between community members of different geographical blocks or genders.

#### 3.6.1. Case Classification by Geographical Block

As noted earlier, the community was divided into four blocks of approximately equal size. Of the twenty-seven interviewees, five resided in Block A, eight resided in Block B, seven resided in Block C, and seven resided in Block D. Overall, there were no notable differences found when the interviews were classified according to their geographical block.

Issues with trash accumulation and lack of trash services were widely noted across all geographical blocks. Trash accumulation was referenced at least 19 times in each block, conveying that this issue impacts all areas of the community to a certain extent. Furthermore, all four blocks had at least 10 references to lack of follow through with trash truck services. This issue was brought up in every interview of each block, with the exception of one interview in Block C.

Most members of the community, regardless of geographical location, were aware of matters relating to their physical health. Physical health was referenced five times in block A, eight times in block B, 12 times in block C and 10 times in block D. This theme was brought up in 80% of block A interviewees, 75% of block B interviewees, 57% of block C interviews and 71% of block D interviewees.

A word reference query was run for each geographical block. These queries listed the top five most frequent words that appeared across all interviews in that specific block. For each of the four blocks, the same three words appeared in the query: “garbage,” “water,” and “come.” Many interviewees expressed that their concerns were felt by the majority of community members and, regardless of demographic differences, everyone in the community felt marginalized to an extent. One interviewee said:

“I mean, so many people around here have nowhere to put their garbage.”

Not only did the interviewees believe that their individual problems were plaguing the majority of the community, they also thought that the solutions to these problems had to come from a larger sphere of influence than the individual. When asked about the role of the government in trash disposal, one interviewee explained:

“The service and the role of the City Council in waste disposal, as it should be, should be for everyone, not only for some part of the population, but it should be for everyone. They should go in all the places, pick up the trash.”

#### 3.6.2. Case Classification by Gender

Of the 27 interviewees, 16 were female and 11 were male. Similar to when interviews were grouped by geographical location, grouping interviews by gender did not produce any distinct results.

Both physical and environmental health implications were noted fairly equally by male and female interviewees in the community. Physical health was referenced 15 times by males and 20 times by females, while environmental health was referenced 10 times by males and 13 times by females.

Both males and females voiced a pressing need for change in their community. Community suggestions were brought up 27 times by men and appeared in all 11 interviews, while women offered suggestions 33 times across 15 out of 16 interviews. Furthermore, community frustration with government was referenced 22 times by male interviewees and 28 times by female interviewees. This theme arose in 81.8% of male interviews and 75% of female interviews.

## 4. Discussion

### 4.1. Interpretation of Results within a Socioecological Framework

By applying a socioecological framework to the results, residents’ perceptions of marginalization and agency were analyzed on four different levels: individual, interpersonal, community, and institutional.

#### 4.1.1. Individual

Residents were quick to acknowledge individual constraints on agency. Most felt that they did not have the ability or resources to fill long-term gaps in public service provision. Because interviewees could not rely on the trash trucks to come, they exercised their limited agency to devise alternative methods of waste disposal. These methods were cited as necessary evils; residents acknowledged the negative health effects, but did not want trash to accumulate in households.

Since many people believed that the causes of the trash problem were out of their control, they rarely suggested solutions that involved individual changes to behavior. The people who dismissed micro-level solutions such as resident-led cleaning initiatives noted the unsustainable nature of these pursuits. One person said that:

“It’s no use cleaning the cañada and picking up the trash...that crap comes from up there [the neighborhood above].”

#### 4.1.2. Interpersonal

At the interpersonal level, concerns for the health of others served to limit certain households’ trash-disposal options. Making reference to vulnerable neighbors and family members, one interviewee noted that:

“There are moments when one cannot burn it [the trash] because then the smoke causes cough. If someone has the flu,... [you can’t] always burn.”

This resident noted that postponing trash burning was not ideal, because the accumulation of garbage attracted mice. On the other hand, most residents who burned trash did not modify their methods to accommodate asthmatic neighbors. This was not for a lack of awareness; rather, constrained agency led to a constant tension between the two goals of waste control and neighborhood health.

#### 4.1.3. Community

In general, residents displayed a deep awareness of the ways in which the community unit experienced pervasive marginalization. They often perceived the sources of their marginalization to be both place-based and political.

A few residents believed that the spatial isolation of the community discouraged companies from providing trash collection services. Many others said that the community’s location beneath another neighborhood made it a magnet for trash run-off. However, when asked about the principal causes of inadequate waste management, interviewees were confident that the primary culprit was political neglect. One resident explained why the lack of government responsiveness could not be attributed to geographic isolation:

“I say that it would be negligence on the part of the politicians as well, because, of course, this is a sector that the politicians should have given work to. Why? Because there are smaller communities, there are more, more isolated, that... [have] been worked on...It’s a little bit of a problem.”

The above quote not only conveys a general frustration with political marginalization, but also a recognition of its influence at the community level. The trash situation was consequently viewed as a community-wide issue instead of a personal problem. One interviewee exemplified this broader consciousness, saying that:

“...I would be willing [to contribute to a solution] because it suits me. I have a lot of help and it’s in my best interest, just as it’s in everyone’s best interest to.”

Because the trash situation was viewed as a community issue, most of the solutions that residents suggested involved collective agency. The neighborhood board was often named as the entity that could best effect change. After providing some ideas for community improvement, one interviewee paused to note that:

“Of course, it would be...according to the community.”

#### 4.1.4. Institutional

As detailed above, residents believed that their inadequate waste management practices were a symptom of government neglect. Interviewees felt that this neglect was not merely a feature of the current administration, but a long-standing, institutionalized pattern. Most people characterized politicians as individuals who did not care about the well-being of community members, although it should be noted that one interviewee did not endorse this narrative on its face. When asked whether the City Council was to blame for trash collection issues, the resident said:

“Yes, if that’s up to the City Council, but here the City Council hires a private company, then that company doesn’t go.”

This interviewee implied that the government’s lack of coordination with its private contractors was a pervasive source of harm. As a whole, residents complained of willful negligence much more frequently than the city’s inability to hold third-parties accountable. However, the above statement illustrates how some people conceptualized sources of marginalization at the institutional level.

In 2019, the government of the Dominican Republic spent 1,296,634.00 $RD or approximately $22,355.76 US dollars on solid waste management services for the entire region [23]. Additionally, $0 dollars were allocated for the sanitation of streets, squares, and parks, and 188,558 $RD or $3251 US dollars were spent on street construction for the southern region [23]. It is important to note that Esfuerzo de Paraíso is not recognized by the government as its own district. Therefore, the local budget only includes the residents of Paraíso that live at the top of the valley region. However, public services such as education and electricity are still provided to residents in Esfuerzo. Furthermore, the city hall “Ayuntamiento de Paraíso’’ outsources labor to private sectors to manage solid waste. In order for trucks to be able to efficiently reach Esfuerzo de Paraíso, concrete roads and a bridge would need to be built as flooding has been an ongoing issue for the community (resident of Esfuerzo, personal communication, 10 January 2020). Theoretically, even if the budget for Paraíso is able to cover these expenses, since the city hall outsources construction to private companies, small budgeted projects are not prioritized by these private companies. Thereby, without proper funding through the federal government of the Dominican Republic, the local government is unable to provide efficient services to the residents of Esfuerzo.

Regardless of whether residents attributed their plight to deliberate, institutionalized negligence or ineffective organizational practices, most agreed that the government had perpetuated the trash problem. It was also widely believed that a solution would have to come at the hands of the government, in the form of trash trucks. The commonly-expressed idea that government was both the cause of the problem and the only solution showed that residents perceived their options to be limited.

### 4.2. Implications

In Esfuerzo de Paraíso, constraints on agency appeared at every level of the socioecological system. Community members were well aware of these overlapping forces, so they were skeptical that individual-level changes would alleviate the trash problem. Residents believed that their individual efficacy was low, and that their best option was to take collective action.

Residents also believed that the most potent marginalizing forces originated at the institutional level. For this reason, many people supported solutions that involved state actors. Community members implied that effective solutions had to address the characteristics of the enabling environment, such as disorganized and unresponsive government. Their sentiments are consistent with past literature, which suggest that sustainable waste management practices must take into account the socioeconomic, institutional and environmental conditions [24].

While residents’ suggestions were valuable, much could also be learned from what was left unsaid. Most agreed that any successful solution had to be initiated at the community level and implemented at the institutional level. Interviewees frequently viewed the local government as the only institution that could implement solutions, even though residents complained of its unreliable nature. Partnerships with non-governmental organizations, non-profit foundations, and international groups were never suggested as possible paths forward. These entities, however, may be able to offer tools that are unavailable at the municipal level. Indeed, a similar community in a nearby informal settlement was able to expand their community-led composting project and better address waste management issues after securing support from a broad coalition of both non-governmental and governmental partners [25]. Residents of Esfuerzo did not consider these alternative sources of support, which could suggest a possible gap in their awareness of available resources. However, the results do not allow for any official conclusions.

Non-traditional trash collection models have produced results in other informal settlements and may help Esfuerzo realize improvements [26,27,28]. In keeping with the sentiments of residents, we propose two possible solutions that are initiated at the community level and implemented with the support of institutional-level actors. The first has proven successful in other locations and involves the combination of community-based microenterprises and international support. In Managua, Nicaragua, community members formed trash collection teams that went house to house [26]. The teams were financed by residential service fees and initially suffered from insufficient funds, but after receiving financial and technical resources from an Italian organization, they were able to maintain and expand their operations [26]. A few settlements in the Dominican Republic have also benefited from similar partnerships. In the Distrito Nacional, community-based microenterprises received resources and advocacy support from a European Union program known as “SABAMAR” or “Saneamiento Ambiental para Barrios Marginales” [27]. With the help of SABAMAR, these microenterprises evolved into recognized community foundations that now contract with the municipal government to collect trash [27]. This model may be a viable solution for Esfuerzo.

If residents of Esfuerzo are unable or unwilling to secure support from international organizations, they may be able to pattern their mobilization after settlements that take advantage of participatory budgeting laws. In Sucre, Venezuela, communities have a say in how the municipal budget is allocated [28]. Residents channeled 2% of funds towards solid waste management and used the money to hire 61 community members who were tasked with daily trash collection [28]. Because the Dominican Congress authorized participatory budgeting at the municipal level in 2007, residents of Esfuerzo could attempt to take action within a similar context [29].

### 4.3. Limitations

It is important to note that while pursuing this study, researchers found some limitations. Most notably, access to a larger range of data would allow us to gather data that are more representative of the entire community. While these findings provide initial analysis on the various implications of waste disposal practices and their effect on community beliefs, the small number of participants has to be approached with caution in regards to generalizing findings. In other geographical and marginalized communities, individuals may experience different difficulties. The households included in this study were from a highly marginalized population that does not represent the larger population of the Dominican Republic.

### 4.4. Future Work

The findings of this interdisciplinary project pertain to a wide range of fields, including environmental health, public health, and community development. Taken together, the results show how a community’s perceptions of marginalization influence the solutions that they consider to be effective. Any organizations who seek to problem solve alongside similarly situated communities can use this work as a starting point to explore the ways that residents prefer to take action. Since sustainable solutions must have the support of the community, such assessments are a necessary precursor to project implementation.

To continue building on this line of work, future studies should seek to determine why marginalized communities recognize some entities as vehicles of change, but not others. Researchers should analyze the resource networks that are available to members of marginalized communities, and ask whether residents are aware of the services offered by different organizations. They should also identify obstacles that prevent disadvantaged populations from utilizing these services or making contact with outside actors.

## 5. Conclusions

This study provides initial empirical research aimed at understanding how community perceptions of marginalization affect short- and long-term agency with regards to waste management. Although no official conclusions can be made from the qualitative research methods of this study, the results highlight the fact that most community members were discontent with their trash disposal practices. Residents reported both physical and environmental health consequences as a result of their subpar waste management. Due to long-standing political, spatial, and economic marginalization, many in the community did not have the means to generate effective solutions at the individual level. They believed that change should be initiated at the community level and implemented with the support of institutional-level actors. Interviewees expressed deep frustration with the government, but generally believed that it was the only institution that could provide them with the necessary resources. Residents almost never suggested solutions that involved non-governmental organizations or community foundations. Future research should explore why marginalized groups may have a limited view of the networks that could offer them support.

## Figures and Tables

**Figure 1 ijerph-18-02872-f001:**
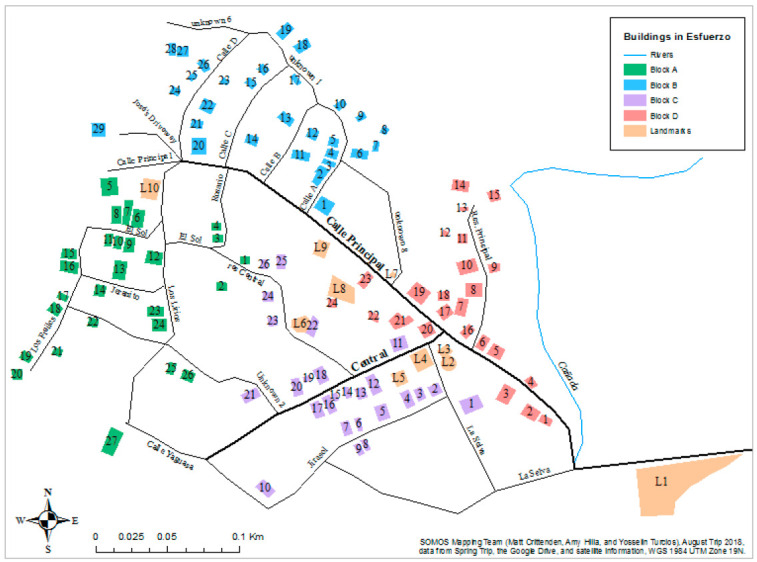
Map of Esfuerzo de Paraíso, Dominican Republic.

**Table 1 ijerph-18-02872-t001:** Semi-Structured Interview Guide Used by Researchers.

Domain	Questions
Waste Management	1. How does trash disposal or waste in general in the community affect health? 2. How do you dispose of trash? Is this how you would ideally want to be disposing of trash? If no, what is stopping you from pursuing that ideal? How do you think *insert trash disposal method here* affects your health? How do you think *insert trash disposal method here* affects the environment? 3. How do most people in the community dispose of trash? Why do most people dispose of trash in this way? 4. What is the role of the local government in waste management? 5. Where does most of the trash in the community come from? 6. How do you think the community can improve trash disposal? 7. If they name a specific solution, ask: Would you be willing to contribute towards that solution?
Other Community Issues	1. Do you have access to sufficient water for drinking, bathing, washing, and other purposes? 2. Do you think the water you have access to is sufficiently clean for drinking? What about bathing, washing clothes, washing dishes? 3. Are you satisfied with your level of access to clean water? If not, how could the community’s water access be improved? 4. Do you view flooding as an issue that impacts your health? If so, how? In what other ways does flooding impact you? 5. Do you see other potential changes or improvements within the community that could further prevent flooding? 6. What other issues affect overall health in the community? 7. What changes would you like to see in your community?

* Researchers adapted the question based on previous answers from interviewees.

**Table 2 ijerph-18-02872-t002:** Summary Statistics by Geographical Block.

Geographical Block	*n*	%
A	5	18.5
B	8	29.6
C	7	25.9
D	7	25.9
Total	27	100

**Table 3 ijerph-18-02872-t003:** Summary Statistics by Gender.

Gender	*n*	%
Male	11	40.7
Female	16	59.2
Total	27	100

## Data Availability

Data sharing is not applicable to this article.

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
