# Peer review of "The Influence of Marginalization on Cultural Attitudes and Trash Disposal Practices in Esfuerzo de Paraíso of the Dominican Republic: A Qualitative Interview Study"

_ijerph, 2021, doi:10.3390/ijerph18062872_

Round 1

Reviewer 1 Report

The reviewer was reading the manuscript with big interesting in people's mind how the see issue of waste management in their daily lives. The paper clearly provides background information, justification of the study, methodology, results and discussion. The reviewer recognises that the paper would be important information for further work to improve the current situation on waste management practices by both citizens and the city hall.

As one comment for consideration, the reviewer was feeling that the paper mainly describes the opinions on waste management from citizens who always complain the city hall. The reviewer recognises that the city hall has a reason why they cannot further invest their resources on waste management in order to respond other major items. 

This is the common among all the cities regardless of income level.

The reviewer would recommend to add discussion on this point.

Reviewer 2 Report

This study provides initial empirical research aimed at understanding how community perceptions of marginalization affect short- and long-term agency with regards to waste management. 

Introduction is quite good. Compare in this section situation and area You describe with some other part of the world

Sufficient methodology for this type of research problem

The writing language is somewhat fictional

I miss the development and proposing a solution to the problem - albeit theoretical. So write some in Your opinion possible solution to solve problem describe in Your manuscript

I would develop the technical sphere with a proposal of technical solutions 

Reviewer 3 Report

I enjoyed reading the article as it provides some attitude and perception of individual in regards to the waste management that is very important. However, the study lacks several important issues:

  1. The interview was done only with the people in one city,  Esfuerzo de Paraíso, in  Dominican Republic. It should not represent the whole country in terms of people social and economic conditions. I would suggest change the title as, "The Influence of Marginalization on Cultural Attitudes and Trash Disposal Practices in Esfuerzo de Paraíso of the Dominican Republic: A Qualitative Interview Study".
  2. The number of interviews (only 27) is not enough to justify with the population even in the city. Open ended questions are not good for statistical grouping and analysis. 
  3. The questions asked during the interview should be listed in the article. Some statistical analysis of data is necessary based on the response to see any significance of the results. 
  4. No conclusion or judgement should be made on the health impact of waste management practices based on open ended interview questions rather than corelating it with the real-word scenario.   

Round 2

Reviewer 3 Report

No further comments/suggestions